# The Cellular and Molecular Immunotherapy in Prostate Cancer

**DOI:** 10.3390/vaccines10081370

**Published:** 2022-08-22

**Authors:** Anirban Goutam Mukherjee, Uddesh Ramesh Wanjari, D. S. Prabakaran, Raja Ganesan, Kaviyarasi Renu, Abhijit Dey, Balachandar Vellingiri, Sabariswaran Kandasamy, Thiyagarajan Ramesh, Abilash Valsala Gopalakrishnan

**Affiliations:** 1Department of Biomedical Sciences, School of Biosciences and Technology, Vellore Institute of Technology (VIT), Vellore 632014, Tamil Nadu, India; 2Department of Radiation Oncology, College of Medicine, Chungbuk National University, Chungdae-ro 1, Seowon-gu, Cheongju 28644, Korea; 3Department of Biotechnology, Ayya Nadar Janaki Ammal College (Autonomous), Srivilliputhur Main Road, Sivakasi 626124, Tamil Nadu, India; 4Institute for Liver and Digestive Diseases, Hallym University, Chuncheon 24252, Korea; 5Centre of Molecular Medicine and Diagnostics (COMManD), Department of Biochemistry, Saveetha Dental College & Hospitals, Saveetha Institute of Medical and Technical Sciences, Saveetha University, Chennai 600077, Tamil Nadu, India; 6Department of Life Sciences, Presidency University, Kolkata 700073, West Bengal, India; 7Human Molecular Cytogenetics and Stem Cell Laboratory, Department of Human Genetics and Molecular Biology, Bharathiar University, Coimbatore 641046, Tamil Nadu, India; 8Water-Energy Nexus Laboratory, Department of Environmental Engineering, University of Seoul, Seoul 02504, Korea; 9Department of Basic Medical Sciences, College of Medicine, Prince Sattam bin Abdulaziz University, P.O. Box 173, Al-Kharj 11942, Saudi Arabia

**Keywords:** prostate cancer, immunotherapy, combination immunotherapy, immune checkpoints, biomarkers

## Abstract

In recent history, immunotherapy has become a viable cancer therapeutic option. However, over many years, its tenets have changed, and it now comprises a range of cancer-focused immunotherapies. Clinical trials are currently looking into monotherapies or combinations of medicines that include immune checkpoint inhibitors (ICI), CART cells, DNA vaccines targeting viruses, and adoptive cellular therapy. According to ongoing studies, the discipline should progress by incorporating patient-tailored immunotherapy, immune checkpoint blockers, other immunotherapeutic medications, hormone therapy, radiotherapy, and chemotherapy. Despite significantly increasing morbidity, immunotherapy can intensify the therapeutic effect and enhance immune responses. The findings for the immunotherapy treatment of advanced prostate cancer (PCa) are compiled in this study, showing that is possible to investigate the current state of immunotherapy, covering new findings, PCa treatment techniques, and research perspectives in the field’s unceasing evolution.

## 1. Introduction

Over the past ten years, there have been many immunotherapy trials for different solid tumors. The developments in cancer immunotherapy go beyond figuring out how the immune system and diseases interact to become indicators of how cancer will progress [1,2,3]. Most treatment for many solid tumors continues to be surgery, followed by chemotherapy and radiation therapy. However, immunotherapy is being increasingly combined with other treatments to increase survival outcomes. Even though immunotherapy seems to hold promise for a variety of solid tumors, prostate cancer (PCa) treatment advancements have been somewhat slow [4]. According to research on the genetic, epidemiologic, and pathophysiologic components of PCa, inflammation is thought to be a key factor at various stages of PCa growth and metastasis [5]. Addressing PCa’s pathogenesis will therefore help in the development of novel combination therapy strategies, with a focus on how the disease responds to various immunomodulatory drugs [4,6,7]. Once a localized illness has been identified, the standard interventional strategy entails radical prostatectomy or radiation therapy, followed by ongoing PSA testing for biochemical recurrence [8]. Chronic inflammation brought on by cellular and genomic destruction by prostatitis is strongly linked to the development and progression of PCa [9]. Epithelial–mesenchymal transition [10,11] and extracellular matrix remodeling brought on by persistent inflammation in the prostate are crucial to the onset and development of the illness. In contrast to other cancers, PCa is a slow-progressing inflammatory disease, making it a prime target for immunotherapy [4]. 

One in seven males in the US will be diagnosed with PCa at some point in their lives, which is particularly common in countries with high human development indexes (HDIs). Prostate-specific antigen (PSA) testing became widely used in the US and Europe in 1986, but it soon fell out of favor given the high rates of false positives, misdiagnosis, and overtreatment. The dramatic growth and decreased PCa mortality can be attributed to these factors [12,13]. PCa accounts for 7.1% of all cancer diagnoses globally, with an expected 1,276,000 reported cases globally in 2018 translating to a 29.3/100,000 men estimated incidence [14]. Northern and Western Europe, Australia/New Zealand (86.4/100,000), and the United States have the maximum rates of PCa [15]. The ASR is 6.3 in countries with low-to-medium HDIs, such as China, India, and Brazil (compared to 11.8 for lungs) [16]. In 2020, 192,000 new instances of PCa were anticipated in the United States, making up 10.6% of all cancer cases. About 12.1% of Americans face a lifetime risk of nearly four times the global risk, and 6.7% of all cancer-related fatalities in men are caused by prostate cancer [17]. Since 1993, when it peaked at 39.3/100,000, the death rate has decreased to 18.9/100,000. Prostate cancer caused 11,714 annual fatalities in the UK from 2015 to 2017, 7% of all cancer-related deaths (superior to the 5.5% in the US) [18]. PCa incidence and mortality have consistently increased in industrialized countries during the past few years. All this advancement was made in the 1980s when PSA screening became widely used and led to the earlier detection and excision of numerous cases of asymptomatic PCa [18,19]. From 2013 to 2017, the five-year survival rate in the UK was 86.6%, as opposed to 97.8% in the US [20]. 

Since Dr. Donald Gleason first described the Gleason score in 1966, it has evolved into a mainstay for managing PCa. The evaluation and treatment of PCa have significantly changed due to the increasing use of PSA screening and needle core biopsy [21]. The Gleason system is applied at various stages of clinical PCa therapy. A key factor in the course of the disease and a criterion for therapy selection is the histological growth patterns of the tumor, as determined by the Gleason grading system. One of five grade groups (GGs) is assigned per the frequency of these patterns [22]. The GG is one of the most crucial prognostic markers for PCa patients and can be used to help choose the treatment course that minimizes the risk of the patient’s progression of the disease [23,24]. Educational initiatives to enhance Gleason grading are advised for all pathologists evaluating prostate biopsies [25]. 

The rigorous description of each growth pattern is arguably the most significant change in PCa treatment. No matter the type of specimen, a Gleason score (GS) of 1 + 1 = 2 should never be given since it has false practical significance [26]. Rarely, if ever, should GS 2–4 be rendered in needle biopsies. The main drawback of using a needle biopsy to diagnose GS 4 is that it cannot be used to see the complete perimeter of the lesion to know whether it is confined. The most common GP found during a prostate needle biopsy is Pattern 3. (NBX). Technically speaking, this pattern is described as distinct, well-formed cancer glands. Since the Gleason grading system’s inception, the concept of GP 4 has changed dramatically [27,28]. The morphological range of GP 5 is broad and comprises comedocarcinoma with central necrosis surrounded by papillary, cribriform, or solid masses, as well as PCa with basically no glandular differentiation, solid tumor sheets, cords, or single cells [21,26]. 

For several solid tumors and hematologic malignancies, therapeutic efforts to engage the immune system versus tumor cells have yielded conflicting outcomes. Nevertheless, immunotherapy has made significant progress over the past ten years and is now a crucial component of the treatment regimen for those with advanced solid malignancies [4,29]. Immunotherapy for people with PCa has historically had minimal efficacy. Immunotherapy is once again considered a potential approach for PCa, particularly castration-resistant prostate cancer (CRPC) [30], to activate anti-tumor immunity due to multiple recent significant discoveries regarding immune processes and new molecular diagnostic platforms [4,12,14]. The discipline must advance with immune checkpoint inhibitors, conventional cytotoxic drugs, and androgen receptor (AR)-targeted medicines in combination with patient-tailored immunotherapy [4,29].

## 2. Epigenetic and Predictive Biomarkers for PCa

### 2.1. DNA Methylation

DNA methylation is one of cancer’s most thoroughly researched epigenetic alterations. CpG sites, or 50-carbon cytosine nucleotides flanked by guanine, are where DNA methylation occurs. In contrast to normal tissues, the DNA methylation pattern is frequently changed in cancer tissues, which results in abnormal gene expression [31]. Gene silences frequently result from DNA methylation, especially in promoter regions. The detection of DNA methylation alterations that (1) may promote tumor initiation and spread or (2) may function as accurate diagnostic, prognostic, or predictive biomarkers is among the objectives of these initiatives [32,33]. Localized prostate tumors show extensive alterations in the hyper- or hypomethylation of certain areas that affect gene expression. As evidence of the possible impact of germline changes on DNA methylation, several methylation differences between tumors have been linked to variations in germline polymorphisms [34]. Zhao et al. also discovered numerous hypo-methylated areas close to the AR gene, along with an enhancer for the gene that had already been discovered close to the promoter and additional sites upstream and downstream of the AR gene [35]. 

Evidence indicates that PCa is entangled with altered methylation patterns. An immunohistochemical analysis of the DNA methylation frequencies in tumorous and typical prostate tissue suggested global hypomethylation in PCa [36]. Santourlidis et al. discovered that the methylation of the LINE-1 was inclined to decline along with increasing tumor stage [37]. These studies were further supported by Schulz et al. 2002, who examined tumor samples and found a connection between DNA hypomethylation, the status of the tumor, and metastasis. A significant link between LINE-1 DNA hypomethylation and chromosomal aberrations in later-stage prostate tumors and lymph node-positive prostate tumors was demonstrated [38,39]. Global DNA hypomethylation might promote malignancy by encouraging genomic instability, whereas specific gene promoter hypomethylation can potentially contribute to the onset and development of malignancy by promoting aberrant gene expression. According to Yegnasubramanian et al., primary prostate tumors and metastatic prostate cancer demonstrate the hypomethylation of CpG islands related to a collection of cancer-testis antigen genes associated with overexpression genes [40].

### 2.2. Histone Modifications

The 30 distinct proteins that make up the histone acetyltransferase family transfer acetyl groups from acetyl-CoA to lysine tails on histones [41]. Pomerantz et al. showed that the acetylation of the lysine 27 site on histone H3 coincides with changes in genome-wide AR binding during the transition from benign to localized tissue castration illness to metastatic CRPC (H3K27ac) [42]. Additionally, identifying different prostate cancer subtypes was aided by incorporating the chromatin immunoprecipitation (ChIP) sequencing of histone modifications, including H3K27Ac data [43].

### 2.3. PD-L1 Expression

The ramifications of PD-L1 expression in prostate cancer are still unclear. Ness et al. assessed the effect of PD-1 expression on intratumoral lymphocytes and PD-L1 expression in tumor epithelial cells in 535 individuals who received prostatectomies [44]. In 92% of the cases, tumor epithelial cells stained with PD-L1 were positive, with 59% having a high PD-L1 intensity score. They discovered a trend toward a poor correlation between tumor epithelial cells that were PD-L1-positive and biochemical failure-free survival. In contrast, they discovered a tendency for individuals with PD-1-positive cells to have lower clinical failure-free survival [45].

### 2.4. Indoleamine-2, 3-Dioxygenase (IDO)

Studies showing that IDO expression was also connected with resistance to anti-CTLA-4 antibodies in mice tumor models demonstrated the importance of IDO as a possible biomarker for the enrolment of cancer patients into immunotherapeutic regimens [46]. As a result, IDO expression may serve as a biomarker for gauging the effectiveness of anti-CTLA-4-targeted therapy. These conclusions support the idea that treating PCa patients with a combination of IDO inhibitors (indoximod), anti-CTLA-4 mAb, and DC-based vaccinations (Provenge) may be a successful course of action [47].

## 3. PCa and the Immune System

“Immunotherapy” refers to a broad category of treatments that use the immune system to fight cancer. Immunotherapy has significantly improved the management of metastatic cancer in the recent years and changed the accepted level of care for several tumor types [48]. It has been difficult to anticipate and comprehend responses across tumor types. While certain metastatic malignancies, such as melanoma, lung cancer, and renal cell carcinoma, have dramatically responded to immunotherapy, PCa has typically failed to demonstrated a meaningful response [49]. Nevertheless, a trivial number of patients with PCa have admirably responded to cells and immunotherapy, indicating that it is worthwhile for future investigation [12].

### Prostate Tumor Microenvironment

An immunosuppressive microenvironment and a “cold” tumor are common descriptions of PCa. By preventing T-effector cell function, tumor-infiltrating lymphocytes (TILs) may aid in the growth of PCa [50]. It has been discovered that TILs from PCa biopsy samples are predisposed to the T regulatory (Treg) and T helper 17 (Th17) phenotypes, which block autoreactive T cells and anti-tumor immune responses [51,52]. Designing treatments that could improve immunological infiltration by antigen-presenting cells (APCs) and effector T cells is of interest [51]. Antigen-presenting dendritic cells (DCs) are crucial for CD8+ T cell activation following tumor eradication. Numerous studies have linked DC tumor infiltration to a better prognosis [53]. It has been demonstrated that androgen deprivation therapy (ADT) induces T cell priming to prostatic antigens and briefly reduces T cell tolerance. These results imply that there may be a synergistic relationship between ADT and immunotherapy [54]. 

The immune system can react to the pathophysiology of neoplasia. Natural killer cells, CD8+ cytotoxic T cells, and macrophages/antigen-presenting cells (APCs) are key cell types used in identifying and eliminating tumor cells [51]. From a pathophysiological perspective, cancers are skilled at creating pathways to stifle immune responses and avoid immune destruction, resulting in evasion and clinical progression [55]. Additionally, tumors can draw in and foster the growth of immune-suppressive cell types, particularly regulatory T cells (Tregs) [54]. Interleukin-10 (IL-10) and transforming growth factor (TGF) are two immunosuppressive substances that tumors may explicitly or implicitly release, helping to create an immunosuppressive microenvironment inside and surrounding the tumor [30,56] (Figure 1).

## 4. Immunotherapy Resistance of PCa

The ineffectiveness of current immunotherapy in men with mPC may be due to a compromised immune system. Cellular immunity in patients with metastatic PCa is defective, and the tumor microenvironment is more immunosuppressive [7,57]. Additionally, myeloid suppressor cells and regulatory T cells exhibit higher inhibitory characteristics in men with mPC in both the bloodstream and the tumor microenvironment [58]. Explanations have been suggested for immunotherapy resistance in prostate cancer, including immunological tolerance due to the slow progression of the disease [59]. Even though this theory is debatable, given that stratified genomic analysis has indicated a greater mutational burden than just that seen in renal cancer, a reduced tumor mutational burden could lead to PCa de novo immunotherapy tolerance. Considering the immunosuppressive prostate TME, it is not easy to generate efficient immunotherapeutics [29,57].

## 5. Immune Checkpoint Inhibition

The FDA has so far authorized the use of the ipilimumab monoclonal antibody as a cancer immunotherapy treatment [60]. Owing to a minimal difference in patient overall survival (OS) between the ipilimumab monotherapy arm and the placebo arm, the initial clinical trial using ipilimumab monotherapy was stopped at phase III [61]. Current clinical trials for mCRPC use mixtures of immune checkpoint inhibitors as a substitute approach. For instance, CheckMate 650, a phase II clinical research study, was started to examine the effects of ipilimumab and nivolumab when given together to mCRPC patients who had become resistant to androgen receptor (AR)-targeted therapy [62,63]. Nevertheless, a new analysis from Cancer Discovery 2019 showed that the two medications together only had a 25% objective response rate [4].

### 5.1. Programmed Cell Death Protein 1 and Programmed Cell Death Protein Ligand-1 (PD-1/PD-L1)

The activation of immune checkpoint pathways, which limit anti-tumor actions by inducing T cell depletion or anergy shown in several forms of solid tumors, is one of the key ways cancer cells avoid immune surveillance [64]. By disrupting T cell co-inhibitory signaling pathways, immune checkpoint inhibitors maintain anti-tumor activity, boosting the tumoricidal effect of the immune system [65]. Avelumab (Bavencio) and atezolizumab are now being researched as anti-PD-L1 immunotherapies that target PD-1 drugs [66]. PD-1 inhibition, which prevents contact with PD-L1 and PD-L2, has been indicated to be a more successful immunotherapeutic method in imposing T cell priming than targeting PD-L1 alone [67].

### 5.2. B7-H3 Blockade

Tumor cells and immature dendritic cells have been shown to express B7-H3, which shares 20–27% amino acid similarity with other B7 family members, though the receptor for B7-H3 is unknown [68]. Nevertheless, excessive B7-H3 expression on cancer cells suppresses T cell activity, which aids immune evasion [69]. Recent research has demonstrated a strong correlation between PCa that expresses B7-H3 and a high Gleason score, mCRPC, and tumor stage [70].

### 5.3. LAG-3

An ICP molecule called LAG-3 (CD223) is present on CD4+ and CD8+ T cells, Treg cells, NK cells, and B cells that have been stimulated [46]. LAG-3 is a potentially effective target for cancer treatment. LAG-3 performs two jobs inside the immune system [71]. Through its association with MHC-II on immature DCs, LAG-3 stimulates DCs and promotes their development. LAG-3 is expressed on T cells and functions as a negative regulatory receptor that directly competes with CD4 for MHCII on APCs [72]. As a result, an inhibitory state is triggered that prevents effector T-cell proliferation and boosts the repressive function of Treg cells. LAG-3 on TILs is bound by anti-LAG-3 mAbs, which prevents LAG-3 from binding to MHCII [73]. The combined targeting of the LAG-3 and PD-1 pathways on anti-cancer efficacy raises the possibility that these two ICPs are co-expressed on anergic T cells and work together to facilitate cancer immune evasion [46].

### 5.4. OX40/OX40L

One other intriguing ICP molecule for cancer-targeted therapy on PCa is OX40 (CD134), a member of the TNFR superfamily. According to research on the expression and activity of OX40 on TILs, activating its signaling may lead to an increase in the effector activity of CD8+ and CD4+ T cells and a reduction in the number of FOXP3+ Treg cells that infiltrate tumors and express OX40 [74]. The growth of T cells, the secretion of cytokines, and the development of memory T cells are increased in immunized tumor-bearing animals when OX40 agonists are administered [75]. The intravenous administration of anti-OX40 and a combination therapy demonstrated tolerable safety. Nine individuals were shown to have 44% lower transitory PSAs and 55% more radiographically stable bone and lymph node metastases throughout an investigation [76].

### 5.5. 4-1BB/4-1BBL

The actions of 4-1BB and 4-1BBL greatly aid the modulation of immune responses. Another TNFR superfamily member, the 4-1BB receptor (CD137), is a co-stimulatory receptor produced on excited CD4+ and CD8+ T cells, active NK cells, and DCs. 4-1BBL expressed on enabled DCs, macrophages, and B cells is bound by 4-1BB [77]. The production of perforin and granzyme is further increased by 4-1BB ligation, enhancing TCR signaling and boosting CD8+ T cell cytotoxicity, IFN-release, and cell growth [78]. Additionally, overexpressing the co-stimulatory molecules CD80 and CD86 quickens the maturation of DCs and boosts the release of IL-6 and IL-12. Combining the anti-CTLA-4 mAb with 4-1BBL-expressing cellular vaccination led to the activation of CTL responses and the shrinkage of existing tumors in PCa-bearing mice [79].

### 5.6. VISTA

A brand-new and potentially effective ICP molecule for PCa immunotherapy, VISTA is also a co-inhibitory ICP that regulates peripheral tolerance. Additionally, VISTA controls the immune response to infections and malignancies in conjunction with co-inhibitory receptors, including PD-1, CTLA-4, LAG-3, and TIM-3 [80]. To prevent the stimulation of T-cell activity in a T-cell-independent way, CD4+ T cells produce VISTA. According to reports, TILs in the TME perform better and anti-VISTA mAbs induce tumor-specific T-cell responses [81,82]. Anti-VISTA monotherapy has also been shown to significantly reduce inducible and transplantable tumors [83,84].

## 6. Immunotherapy for PCa

Over the past ten years, immunotherapy has made significant progress and is now a crucial component of the treatment regimen for patients with advanced solid malignancies. Immunotherapy for people with PCa has historically had minimal efficacy, but immunotherapy is now again considered a PCa treatment option thanks to new scientific discoveries regarding immune systems and cutting-edge molecular diagnostic tools [4,12].

### 6.1. CART Cell Therapy

Autologous cells called chimeric antigen receptor (CAR) T cells are created ex vivo to express a TCR signaling region fused with various antibody regions, allowing them to identify tumor surface antigens without the aid of the MHC [85].

#### 6.1.1. Improving CAR T Cell Persistence

When seeking to destroy highly populated cancer cells in a solid tumor, avoiding CAR T cell depletion is crucial. A high CAR expression can result in increased levels of CAR-mediated tonic signaling in the modified cells, contributing to their exhaustion [86]. CAR expression levels have been shown to significantly impact CAR T cell endurance. A CD22-targeting CAR and an IL-12 transgene were integrated into the TRAC locus and the PD1-encoding PDCD1 locus [85], respectively, in a 2019 study using designer nucleases. Kloss et al. demonstrated that co-transduced T cells only affected cancers displaying both antigens and had no effect on tumors solely expressing one antigen [87].

#### 6.1.2. Specificity and Safety

Two different CARs with two separate scFv domains targeting PSCA or PSMA are injected into the T lymphocyte to prevent or inhibit on-target/off-tumor activity. These CARs either have CD3 signaling domains alone or CD28/4-1BB co-stimulatory domains only in their cytoplasmic tails [88]. The modified T cells are only activated and capable of killing the prostate cancer cells expressing PSCA and PSMA following the simultaneous binding of both CARs to their respective antigens. Off-target effects in cells expressing only one antigen can be prevented this way [89]. Small changes to the CAR design have been demonstrated to significantly affect safety and persistence. A total of 15 residues can be added to the linker sections that join the hinge domain with the scFv or the cytoplasmic tails of a CD19-targeting CAR to significantly alter the character of the resulting CAR T cells and lessen negative consequences [90]. After being infused into CD19+ tumor-bearing animals, these modified CD19 CAR T cells released much less pro-inflammatory cytokines than the prototypical CAR T cells, including IFN and IL-2. The safety of CAR T cells is also being improved through genome editing [91].

### 6.2. Experimental Prostate Cancer Vaccines

#### Aglatimagene Besadenovec

Once triggered by oral valacyclovir, the adenoviral vector aglatimagene besadenovec (ProstAtak; Advantagene Inc.) expressing thymidine kinase (oncolytic virus) can destroy cancer cells. In a randomized (2:1), placebo-controlled phase III trial, aglatimagene besadenovec is being evaluated in individuals with localized illness who are eligible for curative external beam radiation therapy (EBRT; NCT01436968). Aglatimagene besadenovec is also being examined in a different phase II/III trial for patients on active monitoring (NCT02768363) [92,93].

### 6.3. Vaccine-Based Therapies

Prostatic acid phosphatase (PAP) and prostate-specific antigen (PSA) are two immunogenic antigens that are expressed by prostate cancer cells and have both been investigated as potential targets for antigen-based vaccinations [57]. Leukapheresis collects a patient’s dendritic cells, which are then incubated with a PCa-associated antigen at a centralized processing facility before being reinfused into the patient [93]. Antigen-presenting cells (APCs) widely display the GM-CSF receptor. APCs have been demonstrated to cause cytotoxic T-lymphocytes to detect and destroy prostate tumor cells when exposed to PAP in vitro [94].

### 6.4. DNA-Based Vaccines

Current research is focused on DNA vaccines that can take advantage of tumor-associated antigens (TAAs) to build an immune response that results in the strong multiplication and stimulation of tumor antigen-specific T cells [95]. Neoantigen DNA vaccines are now being explored as a customized therapy to encourage T lymphocytes to target tumor cells that express neoantigens. Neoantigen DNA vaccines are created through the whole exome sequencing of tumor and germline DNA to identify tumor-specific antigens [96]. The development of neoantigen DNA vaccines, both dendritic cell and peptide-based, is still in its early phases. Neoantigen DNA vaccines may be effective in treating PCa as well [97].

### 6.5. Cell-Based Vaccines

LNCaP and PC3 are castration-resistant allogenic prostate cancer cell lines that have been genetically modified to overexpress GMCSF, which activates DCs and T cells to produce potent anti-tumor responses [98]. Clinics are not currently testing GM-CSF cellular vaccines for PCa. However, GM-CSF is currently being researched for its application in preclinical testing in various types of PCa vaccines, such as in conjunction with norcantharidin [99].

### 6.6. Peptide-Based Vaccines

In order to activate cytotoxic T lymphocytes (CTLs) and trigger future anti-tumor responses, personalized peptide vaccines (PPVs) use immunization with tumor-specific peptides that can provoke an immune response [100]. Pre-vaccination patient peptides are often screened for their capacity to elicit a CTL or humoral response to the peptides in vitro, which is the conventional technique for selecting peptide candidates for vaccination [101]. Patients with HLA-A24+ PCa have targets such as PAP, PSA, and PSMA. As immunological responses were found in 18 of 21 studied patients, PSA levels decreased in 14 of the 21 patients and 10 of the 21 patients showed no signs of persistent tumors in MRI imaging. The patients usually showed positive treatment activity [102].

### 6.7. Viral Vector-Based Vaccines

Recombinant viral vectors expressing TAA gene sequences are used in viral-based vaccines [103]. A recombinant attenuated vaccinia and fowl pox virus booster engineered to encode TAAs (PSA) and three co-stimulatory proteins—B7-1 (CD80), lymphocyte function-associated antigen 3 (LFA-3) (CD58), and intercellular adhesion molecule-1 (ICAM-1) (CD54)—comprise PROSTVAC (TRICOM), a poxviral-based vaccination regimen [104]. Whether combination therapy with PROSTVAC can improve immune responses is being researched. Another crucial vector employed as an immunological agent to immunize against TAAs is adenovirus 5 (Ad5) [105].

## 7. Combination Immunotherapy

The discipline should advance through immune checkpoint inhibitors, conventional cytotoxic drugs, and androgen receptor (AR)-targeted treatments in conjunction with patient-tailored immunotherapy [4]. Combining these immunological agents or hormone therapy, chemotherapy, radiation, or surgery may be required as the best course of action to enhance clinical results [106]. Several clinical trials evaluating immunotherapeutic combinations are ongoing.

### 7.1. CAR T Cells Combined with Androgen Deprivation Therapy (ADT)

The initial evidence that androgen deprivation has immunomodulatory effects came from studies revealing the enhanced infiltration of CD4+ and CD8+ T cells into prostate tumors and changes in CD4+ T cell differentiation following androgen withdrawal [107]. Research in males with prostate cancer showed that following androgen removal, the number of T cells and androgen-presenting cells, such as dendritic cells and macrophages, increased in prostate tumors [108]. The EphA2 receptor may acquire ligand-independent pro-oncogenic activities that promote cell motility and metastasis but can be inhibited by ligand binding. Therefore, when combined with enzalutamide, EPhA2 may be a promising target antigen for a future CAR T cell therapy against mCRPC [109]. The androgen receptor antagonist flutamide has been shown to increase the cytotoxicity of second-generation CAR T cells directed towards MUC1 [110].

### 7.2. CAR T Cells Combined with Chemotherapy

Chemotherapy aids in releasing cytokines such as IL-2, IL-7, and IL-15, which causes an initial burst of cell proliferation when the CAR T cells are infused [111]. Low-dose chemotherapy may not harm infused CAR T cells according to studies that have shown that some immunosuppressive cells are more sensitive to specific chemotherapeutic drugs than T cells. Overall, chemotherapeutic doses that do not directly affect CAR T cells may increase CAR T cell activity through heightened cytokine release in the TME [112].

### 7.3. Hormone Therapy and Immunotherapy

The immune system is impacted by androgen deprivation therapy because it causes thymic regeneration, which increases the creation of naive T cells, lowers CD4 T-cell tolerance, and increases the number of CD4 effector T cells. Numerous clinical investigations have assessed how castration and immunotherapy interact [93,110].

### 7.4. Chemotherapy and Immunotherapy

Chemotherapeutic medications interfere with the normal cell cycle, which causes tumor cells to either pause their cell cycle or undergo apoptosis [113]. HMGB1, ATP, and CRT are damage-associated molecular patterns (DAMPs) that tumor cells have been shown to emit throughout the ICD process, which can activate local immune cells. Chemotherapy and ICI administration may boost anti-tumor immune activation [114].

### 7.5. Vaccines and Immune Checkpoint Blockades

In 12 chemotherapy-naive mCRPC patients, a phase I trial assessing the combination of ipilimumab and GVAX showed that the combination therapy resulted in a PSA decrease of ≥50 in 25% of patients with few side effects [115]. In another trial of 30 patients with mCRPC, the researchers evaluated the effects of ipilimumab and the PROSTVAC vaccination together. PSA decreases were seen in 14 of the 24 chemotherapy-naive patients, with six individuals experiencing a ≥50% drop [49]. Pembrolizumab and a tumor PAP-specific DNA vaccine were administered to 25 mCRPC patients in research, and it was shown that 11 of the patients experienced increases in PAP-specific CD8 T cells as a result of the therapy. The serum PSA levels of 8 of the 13 individuals who underwent concurrent combination therapy decreased [116,117].

### 7.6. CTLA-4/PD-1 Combination

Anti-PD-1 therapy is expected to activate the immune system’s effector reaction at the level of cancer cells, whereas anti-CTLA-4 therapy is thought to stimulate T cells [118]. Other tumor types are being investigated for combination therapy with ipilimumab and nivolumab, which were shown to result in improved survival (OS) in melanoma. These results served as the impetus for a clinical trial for advanced PCa that combined CTLA-4 blocking and PD-1 inhibition. Ipilimumab and nivolumab are being evaluated in Checkmate 650, a Phase II clinical trial for mCRPC [63] (Figure 2).

## 8. Conclusions

The function and timing of immunotherapy in metastatic prostate cancer are still poorly known. Even though the ongoing immunotherapeutic revolution in medical oncology seems to have ignored prostate cancer, this viewpoint might be shortsighted. According to clinical research, anti-PD-1/PD-L1 and anti-CTLA-4 mAbs greatly enhance the therapeutic effects of cancer vaccines in PCa. Additionally, RT and other chemotherapeutic drugs have been shown to improve anti-cancer efficacy in PCa when CTLA-4 and PD-1/PD-L1 are inhibited. Sipuleucel-T, a therapeutic prostate cancer vaccination, may be responsive to immunotherapeutic techniques according to preliminary evidence from checkpoint inhibitor trials. These treatments will probably comprise a variety of immune-based platforms. One method to maximize immunotherapy in prostate cancer may be to combine medications that can drive T cells to the tumor and transform the “cool” PCa tumor microenvironment into an immunologically “hot” environment. However, the immunosuppressive milieu hampers the establishment of efficient immunotherapies within PCa.

The evidentiary environment for prostate cancer immunotherapy is quickly changing, even though the discovery and approval of immunotherapies in this disease have lagged behind those of other solid tumor malignancies [119]. The cancer treatment paradigm has been completely altered by immunotherapy, making it possible to treat diseases with metastatic spread. The molecular features of prostate cancer have been identified to clarify the factors impeding the beneficial effects of ICIs. Every step in the cancer immunotherapy mechanism, including antigen retrieval, antigen presentation, T cell priming, immune cell homing, reactivating T cells, identifying cancer cells, and carrying out lethal activities, is difficult [120]. According to preliminary findings, combination therapy will probably be required to achieve significant and long-lasting remissions with solid tumor T-cell-redirected therapies. Although preclinical models might be useful in early mechanistic research, these combination tactics will necessitate identifying resistance mechanisms, ideally from patient blood and tissue data. The anticancer effectiveness of CAR-T cell treatments may also require structural adjustments to pass through stromal barriers. Novel immune treatments provide an intriguing new therapeutic option for advanced prostate cancer with the potential for long-lasting, sustainable responses. To overcome the immune-suppressive milieu and physical hurdles specific to prostate cancer, additional adjustments to CAR-T cells or the inclusion of adjuvant medicines may be required [119,121].

## Figures and Tables

**Figure 1 vaccines-10-01370-f001:**
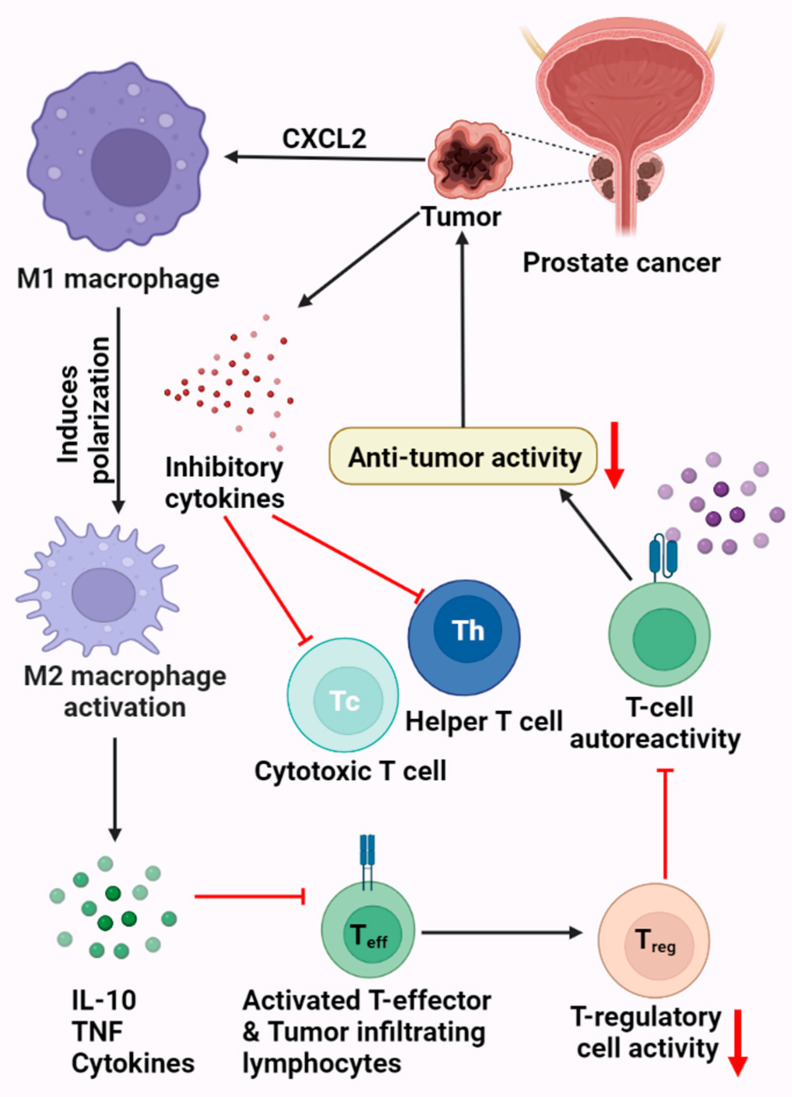
Prostate tumor microenvironment. This figure displays the various mechanistic processes by which PCa can shield or evade the body’s immune system. This includes the production of inhibitory cytokines, resulting in the downregulation of the T_H_ and Tc cells. Prostate cancer cells can also activate the M2 macrophages, leading to the production of IL-10, TNF, and various cytokines, which in turn inhibits the T cells.

**Figure 2 vaccines-10-01370-f002:**
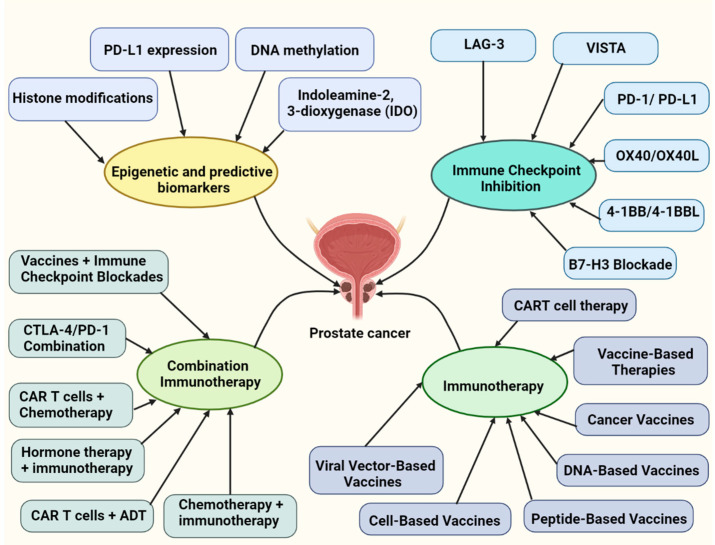
This figure overviews the epigenetic and predictive biomarkers, various immune checkpoint inhibitors, immunotherapies, and combination immunotherapeutic approaches implicated in prostate cancer.

## Data Availability

Data are available from the authors on request (A.V.G.).

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
