# Peer review of "The Cellular and Molecular Immunotherapy in Prostate Cancer"

_vaccines, 2022, doi:10.3390/vaccines10081370_

Round 1
Reviewer 1 Report
The manuscript 'The Cellular and Molecular Immunotherapy in Prostate Cancer' by Mukherjee et al. reviews the current knowledge on Prostate Cancer and treatment strategies.
It address various relevant aspects of Prostate Cancer treatment including immunotherapies / ICI / CART cells and vaccines and therefore represents an valuable contribution to the field.
The manuscript is well written and gives a broad overview on current treatment regimens, however, a short paragraph on the histology of PCa including Gleason Score is missing. The reviewer emphasize to extend the conclusion paragraph and discuss strength and weaknesses / limitation of current strategies in more detail.
Author Response
The manuscript 'The Cellular and Molecular Immunotherapy in Prostate Cancer' by Mukherjee et al. reviews the current knowledge on Prostate Cancer and treatment strategies.
It address various relevant aspects of Prostate Cancer treatment including immunotherapies / ICI / CART cells and vaccines and therefore represents an valuable contribution to the field.
The manuscript is well written and gives a broad overview on current treatment regimens, however, a short paragraph on the histology of PCa including Gleason Score is missing. The reviewer emphasize to extend the conclusion paragraph and discuss strength and weaknesses / limitation of current strategies in more detail.
Response: Authors are thankful to reviewer for their valuable comments. We have added the histology of PCa including Gleason Score in the introduction part (Line number 82-103), and also have extended the conclusion part (Line number 448-465).
Since Dr. Donald Gleason first described the Gleason score in 1966, it has evolved into the mainstay of managing prostate cancer (PCa). The evaluation and treatment of PCa have significantly changed due to the increasing use of PSA screening and needle core biopsy [20]. The Gleason system is applied at various stages of clinical PCa therapy. A key factor in the course of the disease and a criterion for therapy selection is the histological growth patterns of the tumor as determined by the Gleason grading system. One of five Grade Groups (GG) is assigned per the frequency of these patterns [21]. The GG is one of the most crucial prognostic markers for PCa patients and can be used to help choose the treatment course that minimizes the risk of the patient’s progression of the disease [22, 23]. Educational initiatives to enhance Gleason grading are advised for all pathologists evaluating prostate biopsies [24]. The rigorous description of each pattern is arguably the most significant change. No matter the type of specimen, a Gleason score (GS) of 1 + 1 = 2 should never be given since it has false practical significance [25]. Rarely, if ever, should GS 2-4 be rendered in needle biopsies. The main drawback of using a needle biopsy to diagnose GS 4 is that it cannot see the complete perimeter of the lesion to know if it is confined. The most common GP found during a prostate needle biopsy is Pattern 3. (NBX). Technically speaking, it is described as distinct, well-formed cancer glands. Since the Gleason grading system’s inception, the concept of GP 4 has changed dramatically [26, 27]. The morphological range of GP 5 is broad and comprises comedocarcinoma with central necrosis surrounded by papillary, cribriform, or solid masses, PCa with basically no glandular differentiation, solid tumor sheets, cords, or single cells [20, 25].
The evidentiary environment for prostate cancer immunotherapy is quickly changing, even though the discovery and approval of immunotherapies in this disease have lagged behind other solid tumor malignancies [116]. The cancer treatment paradigm has been completely altered by immunotherapy, making it possible to treat diseases with metastatic spread. The molecular features of prostate cancer have been identified to clarify the factors impeding the beneficial effects of ICIs. Every step in the cancer immunotherapy mechanism is difficult, including antigen retrieval, antigen presentation, T cell priming, immune cell homing, reactivating T cells, identifying cancer cells, and carrying out lethal activities [117]. According to preliminary findings, combination therapy will probably be required to achieve significant and long-lasting remissions with solid tumor T-cell-redirected therapies. Although preclinical models might be useful in early mechanistic research, these combination tactics will necessitate identifying resistance mechanisms, ideally from patient blood and tissue data. The anticancer effectiveness of CAR-T cell treatments may also require structural adjustments to get through stromal barriers. Novel immune treatments provide an intriguing new therapeutic option for advanced prostate cancer with the potential for long-lasting, sustainable responses. To overcome the immune-suppressive milieu and physical hurdles specific to prostate cancer, additional adjustments to CAR-T cells or the inclusion of adjuvant medicines may be required [116, 118].

Reviewer 2 Report
The manuscript entitled “The cellular and molecular immunotherapy in prostate cancer” is a review focusing on immunotherapy for prostate cancer. Overall, the manuscript was well organized by the sections. However, most of the references are outdated and not specific to prostate cancer which is a major concern for a review paper.
1) For example, in the second paragraph on page 2, the authors should update the numbers and percentages with references available 2021 at least, instead of 2013-2017.
2) In addition, the references the authors cited are mostly broad and common immunotherapies, but not many references are specific to prostate cancer. For example, in 4.6 VISTA on page 5, I do not find at all whether VISTA was used for the therapy for prostate cancer in research or clinical trials or not.
3) 7. Epigenetic and predictive biomarkers for PC section is out of focus in immunotherapy.
4) The authors need to add one or two illustrations.
Author Response
The manuscript entitled “The cellular and molecular immunotherapy in prostate cancer” is a review focusing on immunotherapy for prostate cancer. Overall, the manuscript was well organized by the sections. However, most of the references are outdated and not specific to prostate cancer which is a major concern for a review paper.
1) For example, in the second paragraph on page 2, the authors should update the numbers and percentages with references available 2021 at least, instead of 2013-2017.
Response 1: Authors are thankful for their valuable comments. We have satisfied the comment by giving the latest references in the second paragraph on page 2. (Line number 63-84)
[12] E. K. Fay and J. N. Graff, "Immunotherapy in prostate cancer," Cancers, vol. 12, no. 7, p. 1752, 2020.
[13] S. J. Ma et al., "Prostate cancer screening patterns among sexual and gender minority individuals," European Urology, vol. 79, no. 5, pp. 588-592, 2021.
[14] H. Westdorp et al., "Immunotherapy for prostate cancer: lessons from responses to tumor-associated antigens," Frontiers in immunology, vol. 5, p. 191, 2014.
[15] M. M. Al-Hashimi, "Trends in breast cancer incidence in Iraq during the period 2000-2019," Asian Pacific journal of cancer prevention: APJCP, vol. 22, no. 12, p. 3889, 2021.
[16] P. Rawla, T. Sunkara, and A. Barsouk, "Epidemiology of colorectal cancer: incidence, mortality, survival, and risk factors," Gastroenterology Review/PrzeglÄ…d Gastroenterologiczny, vol. 14, no. 2, pp. 89-103, 2019.
[17] M. B. Culp, I. Soerjomataram, J. A. Efstathiou, F. Bray, and A. Jemal, "Recent global patterns in prostate cancer incidence and mortality rates," European urology, vol. 77, no. 1, pp. 38-52, 2020.
[18] A. Barsouk et al., "Epidemiology, staging and management of prostate cancer," Medical Sciences, vol. 8, no. 3, p. 28, 2020.
[19] T. Pathirana et al., "Trends in Prostate Specific Antigen (PSA) testing and prostate cancer incidence and mortality in Australia: A critical analysis," Cancer Epidemiology, vol. 77, p. 102093, 2022.
[20] E. Cheng et al., "Long-term survival and causes of death after diagnoses of common cancers in 3 cohorts of US health professionals," JNCI cancer spectrum, vol. 6, no. 2, p. pkac021, 2022.
2) In addition, the references the authors cited are mostly broad and common immunotherapies, but not many references are specific to prostate cancer. For example, in 4.6 VISTA on page 5, I do not find at all whether VISTA was used for the therapy for prostate cancer in research or clinical trials or not.
Response 2: We have updated the references as per the comments (Line Number 289-296)
[80] N. Li et al., "Immune-checkpoint protein VISTA critically regulates the IL-23/IL-17 inflammatory axis," Scientific reports, vol. 7, no. 1, pp. 1-11, 2017.
[81] I. Le Mercier et al., "VISTA regulates the development of protective antitumor immunity," Cancer research, vol. 74, no. 7, pp. 1933-1944, 2014.
[82] M. Tagliamento et al., "VISTA: A Promising Target for Cancer Immunotherapy?," ImmunoTargets Therapy, vol. 10, p. 185, 2021.
[83] E. C. Nowak et al., "Immunoregulatory functions of VISTA," Immunological reviews, vol. 276, no. 1, pp. 66-79, 2017.
[84] L. Yuan, J. Tatineni, K. M. Mahoney, and G. J. Freeman, "VISTA: a mediator of quiescence and a promising target in cancer immunotherapy," Trends in immunology, vol. 42, no. 3, pp. 209-227, 2021.
3) 7. Epigenetic and predictive biomarkers for PC section is out of focus in immunotherapy.
Response 3: Authors are thankful the reviewer for this comment. We have rearranged the position of Epigenetic and predictive biomarkers for PC section in the exact section where it is required in order to solve this issue.
4) The authors need to add one or two illustrations.
Response 4: We have included the figures in the main text are as
Figure 1: Prostate tumor microenvironment. This figure displays the various mechanistic processes by which PCa can shield or evade the body’s immune system. This includes the production of inhibitory cytokines resulting in the downregulation of the TH and Tc cells and blocking B cell and NK cells. Prostate cancer cells can also activate the M2 macrophages, leading to the production of IL-10, TNF, and various cytokines, which in turn inhibits the T cell.
Figure 2: This figure overviews the epigenetic and predictive biomarkers, various immune checkpoint inhibitors, immunotherapy, and combination immunotherapeutic approaches implicated in prostate cancer.

Reviewer 3 Report
This manuscript by Mukherjee et al. summarizes the cellular and molecular immunotherapy in prostate cancer. This is a comprehensive review, I recommend a minor revision.
1. More background should be included in INTRODUCTION to enrich it.
2. Some representative figures should be added in the main text.
3. Authors should provide their own perspectives toward this research area.
Author Response
This manuscript by Mukherjee et al. summarizes the cellular and molecular immunotherapy in prostate cancer. This is a comprehensive review, I recommend a minor revision.
- More background should be included in INTRODUCTION to enrich it.
Response 1: Authors are thankful to the reviewer for their valuable comments. We have made the changes and have extended Introduction part (Line number 82-103).
Since Dr. Donald Gleason first described the Gleason score in 1966, it has evolved into the mainstay of managing prostate cancer (PCa). The evaluation and treatment of PCa have significantly changed due to the increasing use of PSA screening and needle core biopsy [20]. The Gleason system is applied at various stages of clinical PCa therapy. A key factor in the course of the disease and a criterion for therapy selection is the histological growth patterns of the tumor as determined by the Gleason grading system. One of five Grade Groups (GG) is assigned per the frequency of these patterns [21]. The GG is one of the most crucial prognostic markers for PCa patients and can be used to help choose the treatment course that minimizes the risk of the patient’s progression of the disease [22, 23]. Educational initiatives to enhance Gleason grading are advised for all pathologists evaluating prostate biopsies [24]. The rigorous description of each pattern is arguably the most significant change. No matter the type of specimen, a Gleason score (GS) of 1 + 1 = 2 should never be given since it has false practical significance [25]. Rarely, if ever, should GS 2-4 be rendered in needle biopsies. The main drawback of using a needle biopsy to diagnose GS 4 is that it cannot see the complete perimeter of the lesion to know if it is confined. The most common GP found during a prostate needle biopsy is Pattern 3. (NBX). Technically speaking, it is described as distinct, well-formed cancer glands. Since the Gleason grading system’s inception, the concept of GP 4 has changed dramatically [26, 27]. The morphological range of GP 5 is broad and comprises comedocarcinoma with central necrosis surrounded by papillary, cribriform, or solid masses, PCa with basically no glandular differentiation, solid tumor sheets, cords, or single cells [20, 25].
- Some representative figures should be added in the main text.
Response 2: We have included the figures in the main text are as
Figure 1: Prostate tumor microenvironment. This figure displays the various mechanistic processes by which prostate cancer can shield or evade the body’s immune system. This includes the production of inhibitory cytokines resulting in the downregulation of the TH and Tc cells and blocking B cell and NK cells. Prostate cancer cells can also activate the M2 macrophages, leading to the production of IL-10, TNF, and various cytokines, which in turn inhibits the T cell.
Figure 2: This figure overviews the epigenetic and predictive biomarkers, various immune checkpoint inhibitors, immunotherapy, and combination immunotherapeutic approaches implicated in prostate cancer.
- Authors should provide their own perspectives toward this research area.
Response 3: We have satisfied the comment in the conclusion part (Line number 448-465).
The evidentiary environment for prostate cancer immunotherapy is quickly changing, even though the discovery and approval of immunotherapies in this disease have lagged behind other solid tumor malignancies [116]. The cancer treatment paradigm has been completely altered by immunotherapy, making it possible to treat diseases with metastatic spread. The molecular features of prostate cancer have been identified to clarify the factors impeding the beneficial effects of ICIs. Every step in the cancer immunotherapy mechanism is difficult, including antigen retrieval, antigen presentation, T cell priming, immune cell homing, reactivating T cells, identifying cancer cells, and carrying out lethal activities [117]. According to preliminary findings, combination therapy will probably be required to achieve significant and long-lasting remissions with solid tumor T-cell-redirected therapies. Although preclinical models might be useful in early mechanistic research, these combination tactics will necessitate identifying resistance mechanisms, ideally from patient blood and tissue data. The anticancer effectiveness of CAR-T cell treatments may also require structural adjustments to get through stromal barriers. Novel immune treatments provide an intriguing new therapeutic option for advanced prostate cancer with the potential for long-lasting, sustainable responses. To overcome the immune-suppressive milieu and physical hurdles specific to prostate cancer, additional adjustments to CAR-T cells or the inclusion of adjuvant medicines may be required [116, 118].

Round 2
Reviewer 2 Report
The manuscript was very much improved by adding the figures and updating the references.